# Endogenously Produced SARS-CoV-2 Specific IgG Antibodies May Have a Limited Impact on Clearing Nasal Shedding of Virus during Primary Infection in Humans

**DOI:** 10.3390/v13030516

**Published:** 2021-03-20

**Authors:** Shuyi Yang, Keith R. Jerome, Alexander L. Greninger, Joshua T. Schiffer, Ashish Goyal

**Affiliations:** 1Department of Data Science, University of California San Diego, La Jolla, CA 92093, USA; s7yang@ucsd.edu; 2Vaccine and Infectious Disease Division, Fred Hutchinson Cancer Research Center, Seattle, WA 98109, USA; kjerome@fredhutch.org (K.R.J.); agrening@uw.edu (A.L.G.); 3Department of Laboratory Medicine and Pathology, University of Washington School of Medicine, Seattle, WA 98195, USA; 4Clinical Research Division, Fred Hutchinson Cancer Research Center, Seattle, WA 98910, USA; 5Department of Medicine, University of Washington, Seattle, WA 98195, USA

**Keywords:** SARS-CoV-2, IgG antibodies, IgM antibodies, severity, mathematical model

## Abstract

While SARS-CoV-2 specific neutralizing antibodies have been developed for therapeutic purposes, the specific viral triggers that drive the generation of SARS-CoV-2 specific IgG and IgM antibodies remain only partially characterized. Moreover, it is unknown whether endogenously derived antibodies drive viral clearance that might result in mitigation of clinical severity during natural infection. We developed a series of non-linear mathematical models to investigate whether SARS-CoV-2 viral and antibody kinetics are coupled or governed by separate processes. Patients with severe disease had a higher production rate of IgG but not IgM antibodies. Maximal levels of both isotypes were governed by their production rate rather than different saturation levels between people. Our results suggest that an exponential surge in IgG levels occurs approximately 5–10 days after symptom onset with no requirement for continual antigenic stimulation. SARS-CoV-2 specific IgG antibodies appear to have limited to no effect on viral dynamics but may enhance viral clearance late during primary infection resulting from the binding effect of antibody to virus, rather than neutralization. In conclusion, SARS-CoV-2 specific IgG antibodies may play only a limited role in clearing infection from the nasal passages despite providing long-term immunity against infection following vaccination or prior infection.

## 1. Introduction

The interplay between the SARS-CoV-2 life cycle and humoral immune responses during natural infection has only been partially described. While neutralizing antibodies are detected 1–2 weeks after symptom onset, it is less certain whether these antibodies are relevant for clearance of the virus or whether they only protect viral re-exposure. A better understanding of viral antibody interactions is critical to specify the optimal timing for neutralizing antibody treatments which may be more effective according to the stage of infection [1,2,3,4,5,6,7]. Indeed, recent trials of neutralizing antibody infusions demonstrate antiviral and clinical efficacy though only if dosed early in infection prior to clinical decompensation [8,9,10]. A deeper understanding of the link between viral loads, antibody dynamics, and COVID-19 severity may also assist in the identification of immune surrogates of protection following infection or vaccination. The specific role of individual antibody isotypes is also not well understood [5,11,12,13].

The present study aims to define some of these basic features of humoral antibody responses. Using mathematical modeling and longitudinal datasets of IgG, IgM, and viral loads, we explored (i) how SARS-CoV-2 viral loads relate to the generation of antibodies, (ii) how and when different isotypes of antibodies correlate with each other, and (iii) whether SARS-CoV-2 antibody levels alter viral load dynamics.

## 2. Materials and Methods

### 2.1. The Raw Data

Three groups of longitudinal datasets were employed for modeling purposes. The first dataset provided IgM and IgG levels from 26 hospitalized SARS-CoV-2 patients in China, and 6 of them developed severe symptoms characterized as the development of dyspnea and/or hypoxemia and rapid progression to acute respiratory distress syndrome, septic shock, refractory metabolic acidosis, coagulopathy, or multiple organ failure [14]. The IgG and IgM antibody levels were measured using the magnetic chemiluminescence immunoassay (MCLIA) value divided by the cutoff (absorbance/cut off, S/CO) with peptides derived from the amino acid sequence of ORF1a/b, spike (S) protein, and nucleocapsid (N) protein [15].

We digitized the second dataset that captured anti-nucleocapsid IgG antibodies and SARS-CoV-2 viral loads in 6 hospitalized patients in Washington State, USA [16]. IgG antibodies were measured using Abbott Architect anti-SARS-CoV-2 nucleocapsid IgG index value, and the cut-off for seropositivity suggested by the manufacturer was 1.40 [17]. The viral loads from nasopharyngeal swabs were originally measured as cycle threshold values with the detection limit 40 using SARS-CoV-2 qRT-PCR test applying Hologic Panther Fusion and laboratory-developed test (LDT) assays.

Finally, the third dataset was measured from two patients who developed mild symptoms of COVID-19. The digitization yielded SARS-CoV-2 viral loads and IgG antibodies against another target site S1 protein of SARS-CoV-2 [18]. Antibodies were measured using optical density value with the cut-off to be 1.1. The viral loads were measured as average cycle threshold values through nasopharyngeal swabs tested by RT-qPCR that applied the TaqMan SARS-CoV-2 Assay Kit v2, the 2019-nCoV CDC qPCR Probe Assay, or the Roche Cobas SARS-CoV-2 Test CE-IVD. The cycle threshold detection limit was 40 [18]. If unavailable, we converted the cycle threshold values into copies/mL using the relation reported in [19].

### 2.2. Mathematical Model Recapitulating Antibody Kinetics

We introduce a simple model that recapitulates IgG and IgM kinetics (Model M3, Appendix A) as follows:(1)$$\frac{{\mathrm{dI}}_{M}}{\mathrm{dt}}=\frac{r_{M}I_{M}}{k_{M}+I_{M}}-d_{M}I_{M}\;\frac{{\mathrm{dI}}_{G}}{\mathrm{dt}}=\frac{r_{G}I_{G}}{k_{G}+I_{G}}-d_{G}I_{G}$$

In this model, $r_{M},{\;k}_{M},{\;r}_{G},$ and $k_{G}$ represent the production rate of IgM, the level of IgM antibodies at which its production rate becomes 50%, the production rate of IgG, the level of IgG antibodies at which its production rate becomes 50%, respectively. The model is derived from the complex dynamics of the development of humoral immunity involving B cells (Appendix A, [20,21,22,23]) and this derivation is provided in Appendix A.

Two variants of the model (1) were also employed for fitting purposes. This included, (i) IgG and IgM production rates that are independent of antibody concentration ($k_{G}=0$ and $k_{M}=0$, Model M1 in Appendix A) and (ii) non-saturated proliferation of different B cell subsets leading to unregulated IgG and IgM production ($k_{G}+I_{G}=1$ and $k_{M}+I_{M}=1$, Model M2 in Appendix A) [24].

### 2.3. Mathematical Model Recapitulating the Interplay between Antibodies and SARS-CoV-2 Viral Loads

To recapitulate longitudinal viral loads in addition to antibody data, we extended models in the previous section. These extended models mainly consisted of two compartments. The first compartment corresponds to the first 3 equations below in model (2). It recapitulates SARS-CoV-2 viral loads over time and is inspired from [25]. This model assumes that SARS-CoV-2 (V) infects susceptible cells (T) at rate β and converts them to infected cells (I). Infected cells are cleared by innate immune responses in a density-dependent manner at a rate ${\delta \mathrm{II}}^{k}$. More SARS-CoV-2 viruses are produced by infected cells at rate p, and rate c represents the natural clearance. The second compartment corresponds to the last equation and is borrowed from the best model M3 in Appendix A. After the first $\tau_{G}$ days, the IgG level ($I_{G}$) begins to grow at rate $r_{G}$ and becomes saturated at high levels of IgG antibodies (modeled by the term $k_{G}$). The natural clearance rate of IgG is denoted by $d_{G}$. The model is thus given by,
(2)$$\frac{\mathrm{dT}}{\mathrm{dt}}=-\beta \mathrm{VT}\;\frac{\mathrm{dI}}{\mathrm{dt}}=\beta \mathrm{VT}-{\delta \mathrm{II}}^{k}\;\frac{\mathrm{dV}}{\mathrm{dt}}=\mathrm{pI}-\mathrm{cV}\;\frac{{\mathrm{dI}}_{G}}{\mathrm{dt}}=\frac{r_{G}I_{G}}{k_{G}+I_{G}}-d_{G}I_{G}$$

To investigate the role of IgG antibodies on viral loads, we modified model (2) by assuming that antibodies have binding effects, neutralizing effects, or effects on clearing infected cells. This is shown in model (3), where the neutralizing efficacy is represented by the term $\frac{I_{G}}{k_{E}+I_{G}}$, which suggests higher levels of IgG antibody give rise to higher neutralizing efficacy and prevent viral infections. Here, parameter $k_{E}$ represents the value of IgG antibodies at which the neutralization efficacy becomes 50%. The binding effect, which is responsible for the faster clearance of the virus, is captured by the term ${\nu I}_{G}V$. Term $\phi {\mathrm{II}}_{G}$ shows the possible effect of antibodies on removing infected cells. The model with the included effects of antibodies is thus given by,
(3)$$\frac{\mathrm{dT}}{\mathrm{dt}}=-\beta \left(1-\frac{I_{G}}{k_{E}+I_{G}}\right)\mathrm{VT}\;\frac{\mathrm{dI}}{\mathrm{dt}}=\beta \left(1-\frac{I_{G}}{k_{E}+I_{G}}\right)\mathrm{VT}-{\delta \mathrm{II}}^{k}-\phi {\mathrm{II}}_{G}\;\frac{\mathrm{dV}}{\mathrm{dt}}=\mathrm{pI}-\mathrm{cV}-{\nu I}_{G}V\;\frac{{\mathrm{dI}}_{G}}{\mathrm{dt}}=\frac{r_{G}I_{G}}{k_{G}+I_{G}}-d_{G}I_{G}$$

In total, we modeled 38 competing hypotheses assuming: (i) antibodies have no effects, only binding effects, only neutralizing effects or both (models with prefix MP, MQ, MR, and MS in Appendix A, respectively); (ii) there is one-off stimulation, continuous stimulation, or delayed-continuous stimulation of B cells for antibody production (models with suffix v1–v3, v4–v6, v7–v9 in Appendix A, respectively); and, (iii) non-saturated antibody production, saturation by high levels of IgG, or saturation by high levels of viral loads (models with suffix v1–v4–v7, v2–v5–v8, v3–v6–v9 in Appendix A, respectively). In addition, we also fit two additional models that assume that antibodies facilitate the death of infected cells (models MT-v1 and MT-v2 in Appendix A).

### 2.4. Fitting Procedure

On a given dataset, we fit models with competing hypotheses as listed in Appendix A at the population level using nonlinear mixed-effects modeling in Monolix 2019R2 (www.lixoft.eu, accessed: 9 February 2021). Under this population-level fitting approach, the value of an unknown parameter for an individual is assumed to be drawn from a distribution with a fixed value (capturing inter-individual similarity) and a standard deviation of the random effects (capturing inter-individual variability). Then the fixed and/or the standard deviation of the random effects for all unknown parameters in our models were estimated using the Stochastic Approximation Expectation-Maximization (SAEM) algorithm [26].

The model selection was based on the Akaike Information Criteria (AIC) [27]. We calculated AIC as −2LL+2m, where m represents the number of unknown parameters in the model and LL denotes log-likelihood. Models with lower AIC are favored and if the difference is more than 2, then there is strong support for the model with lower AIC by the experimental data [27].

For sensitivity analysis, we employed Latin Hypercube Sampling and Partial Rank Correlation Coefficient (LHS-PRCC). We utilized the Wilcoxon rank-sum test to compare the differences in estimated parameter values between severe and non-severe patients while using Pearson correlation to compute the correlation between two variables. Finally, we employed linear-mixed models to test if levels of a specific antibody at any time point in one severity group are higher than another. As each group was inclusive of multiple patients, we nested patients within their respective groups in the linear-mixed model. Specifically, for each individual antibody, we employed linear mixed-effects model “Ab~time + (1 + time|Severity/PID)”, where Ab is the antibody level on log2 scale, time is the days since the onset of symptoms, severity is the severity index (0: non-severe; 1: severe) of the patient with patient ID (PID) nested. The analysis was performed using the package “lme4” and “lmerTest” in the software R.

## 3. Results

### 3.1. Longitudinal IgG and IgM Dynamics

IgG and IgM antibody levels were recorded up to 35 days after the onset of symptoms in 26 hospitalized SARS-CoV-2 patients in China; six of these patients developed severe symptoms [14]. IgM and IgG levels were significantly higher in severe patients compared to non-severe patients (0.0004 and 9.5 × 10^−8^, linear mixed-effects model) (Appendix A). Common characteristics in this dataset included an exponential increase in both antibodies after some delay followed by a plateau towards steady-state levels (Appendix A). The IgG and IgM antibody levels were measured using the magnetic chemiluminescence immunoassay (MCLIA) values divided by the cutoff (absorbance/cut off, SCO) with peptides derived from the amino acid sequence of ORF1a/b, spike (S) protein, and nucleocapsid (N) protein [15].

### 3.2. Mathematical Model of IgG and IgM Dynamics

To capture the observed antibody expansion dynamics, we developed a mathematical model based on the process of development of humoral immunity involving B cells (Figure 1A). The model includes the density-dependent production of IgM and IgG antibodies at rates $r_{M}$ and $r_{G}$, respectively, after a delay of $\tau_{M}$ and $\tau_{G}$, respectively (model M3 in Appendix A). The delay terms capture the lag caused due to initial phases of the development of humoral immunity, which could not be explicitly modeled due to the lack of data. Moreover, the production of antibodies is assumed to become saturated at high levels of the corresponding antibody (regulated by parameters $k_{M}$ and $k_{G}$) whereas IgM and IgG antibodies are naturally cleared at rates $d_{M}$ and $d_{G}$, respectively. This model is a simplification of a complete and fully descriptive model of the development of humoral immunity as shown in Appendix A and the process of reduction is shown in Material and Methods.

The model described above was best supported by the data over alternative models which assumed that the production rate of antibodies either occurs at a constant rate (model M1, Appendix A) or remains unsaturated (model M2, Appendix A). The fits under the best model are shown in Figure 1B using estimated parameter values in Appendix A.

### 3.3. Differences between SARS-CoV-2 IgG and IgM Production Rates

The model analysis revealed that it takes ~0.6 more days on average from symptoms onset to the production of IgM (median~9.1 days, range: 3.0–16.3 days) than IgG antibodies (median~8.5 days, range: 3.4–20.0 days) at rates 3.22 SCO (signal-to-cutoff ratio)/day (range: 0.25–22.62 SCO/day) and 8.94 SCO/day (range: 0.66–71.69 SCO/day), respectively (*p* = 0.04, Wilcoxon rank-sum test) (Appendix A). IgM and IgG production rates become half-saturated at 7.4 × 10^−5^ SCO and 1.3 SCO, respectively. Modeling further yields that IgG antibodies have a longer half-life (~2.7 days) compared to IgM antibodies (~1.4 days). Finally, production rates of IgG production are considerably higher than IgM antibodies.

### 3.4. SARS-CoV-2 Antibody Kinetics during Non-Severe Versus Severe Infection

The time-delay between symptom onset and the start of production of IgM and IgG antibodies in response to infection ($\tau_{M}$ and $\tau_{G}$, *p* = 0.84 and *p* = 0.79, Wilcoxon rank-sum test) as well as the baseline concentration of IgM and IgG antibodies ($I_{M0}$ and $I_{G0}$, *p* = 0.44 and *p* = 0.13, Wilcoxon rank-sum test) were not statistically different between two groups (Figure 2A). Only the production rate of IgG antibodies ($r_{G}$) was significantly larger in severe patients compared to non-severe patients (*p* = 0.04, Wilcoxon rank-sum test) while the production rate of IgM antibodies ($r_{M}$) was not significantly higher in severe patients (*p* = 0.08, Wilcoxon rank-sum test). We believe that this could be because IgM responses are more general than IgG responses as the latter are produced with the help of T cells.

We next performed a sensitivity analysis using Latin Hypercube Sampling and Partial Rank Correlation Coefficient (LHS-PRCC) on all the parameters of our best model that had been reduced to 2-dimension (Figure 2B). We found that the production rates of IgM and IgG antibodies ($r_{M}$ and $r_{G}$) had a highly positively correlated (r~1) impact on peak levels of IgM and IgG antibodies. The baseline concentration of antibodies also positively impacted antibody peak levels, suggesting that pre-existing immunity could accelerate antibody response and prevent re-infection. The delay before the onset of antibody production had almost no impact on peak levels of antibodies (Figure 2B).

### 3.5. Mathematical Model of SARS-CoV-2 Viral Load and Nucleocapsid IgG Dynamics in 6 Hospitalized Patients

Both nucleocapsid IgG antibodies and SARS-CoV-2 viral loads were recorded in six hospitalized patients in Washington State [16] (Appendix A). To reproduce the observed dynamics, we tested 38 competing mathematical models which differ according to mechanisms by which antibodies can be produced in response to the SARS-CoV-2 antigen and the subsequent effect of IgG antibodies on SARS-CoV-2 viral loads (Figure 3). We explored the effect of viral loads on antibody production and tested models with competing hypotheses including, (i) the presence of viral antigen early on during infection “once” triggers B cell stimulation leading to the programmed production of antibodies (termed one-off stimulation, marked v1–v3 in Appendix A), (ii) the viral antigen levels throughout the course of infection "continuously" triggers B cell stimulation in a density-dependent manner and the antibodies are produced in response to the expansion or contraction of viral loads (termed continuous stimulation, marked v4–v6 in Appendix A), and (iii) the antibodies are produced “continuously” in response to the expansion or contraction of viral loads but in a delayed manner (termed delayed-continuous stimulation, marked v7–v9 in Appendix A). We also explored the effect of antibodies on virus assuming no effect, binding effects, neutralization effects, and both binding and neutralization effects, which are represented by models marked as MP, MQ, MR, and MS, respectively in Appendix A. Neutralization is assumed to prevent viral entry onto cells whereas binding clears the virus more rapidly from the nasal passages. Furthermore, we investigated if antibodies exhibit “antibody-dependent clearance of infected cells [28]” (models marked as MT in Appendix A).

The best-performing model was comprised of two components. The first component addresses the viral dynamics as captured in our prior modeling [25]. In this model, SARS-CoV-2 (V) infects susceptible cells (T) with infectivity β and converts them to infected cells (I). Infected cells are cleared by an innate response in a density-dependent manner at rate ${\delta \mathrm{II}}^{k}$. More SARS-CoV-2 viruses are produced by infected cells at rate p, and rate c indirectly stands for all the other immune responses that are not directly related to IgG antibody but still help clear the SARS-CoV-2 viruses. Our model captured viral dynamics well and predicted that viral RNA concentrations peak 2.3 days (min, max: 1.8, 3.4) after the first detection, in line with the literature [29].

The dynamics of IgG were best captured by the model that also recapitulated the antibody data in the previous section (i.e., M3 from Appendix A) suggesting that there is one-off stimulation of B cells leading to the programmed production of antibodies in a self-saturating manner. Despite being one-off stimulation, the delay before the IgG antibodies appear is strongly correlated with the clearance rate of SARS-CoV-2 infected cells (r = −0.995, Pearson Correlation, Appendix A).

The best performing model further suggested that IgG antibodies exhibit no effects on the viral loads (model MP-v2 in Appendix A, AIC = 196). Though the inclusion of antibody levels into the model does not improve fit to the data, the experimental data support that IgG antibodies are more likely to lead to a faster clearance of SARS-CoV-2 viral loads through binding effects rather than neutralization effects (AIC = 208 with MQ-v2 vs. AIC = 214.1 with MR-v2, Appendix A). The timing of the decrease in viral loads also supports our modeling finding as binding effects but not the neutralization effects of IgG antibodies would allow viral loads to decrease simultaneously as antibodies emerge. This is because neutralization effects result in the inhibition of new infections which will result in a delay since an infected cell will still have sufficient time to produce a full round of viral progeny [30].

The corresponding fits are shown in Figure 4 and the individual parameters estimates are provided in Appendix A. Of note, other models that performed nearly as well all included one-off stimulation of antibody production and saturating levels of IgG meaning that these mechanisms were most critical to fit the data.

We further exhibit that IgG binding effects may only play a minor role in clearing SARS-CoV-2 nasal viral loads during primary infection as our simulations suggest that even in the absence of anti-SARS-CoV-2 IgG, the clearance would have been achieved in no more than 5 days after the observed day of viral clearance in all patients except P5. In fact, when binding effects are assumed, the effect of IgG only becomes prominent on SARS-CoV-2 viral loads approximately 7 days after the day of hospitalization in almost all patients except P5 (Figure 5). This is because IgG antibodies are detectable only after ~7 days from the day of hospitalization (or, ~15 days after the infection, Appendix A). Simulations further suggest that IgG antibodies with strong binding effects could accelerate the clearance of SARS-CoV-2 viral loads in those patients who do not naturally clear infection (patient P5 in Figure 5) and thus points towards the potential use of antibodies in the treatment of SARS-CoV-2, especially if given during the early phase of infection, preferably before the viral peak is achieved (Figure 5).

Next, we aimed to determine differences in the dynamics of anti-spike and anti-nucleocapsid IgG and their impact on viral dynamics. For this purpose, we digitized anti-spike IgG and SARS-CoV-2 viral loads from two mild cases of SARS-CoV-2 in Switzerland [18]. The best performing model from Appendix A also recapitulated the additional data from 2 patients (Figure 6A) using estimated parameters in Appendix A. Our analysis reveals that anti-spike and anti-nucleocapsid IgG have similar effects on SARS-CoV-2 viral loads (i.e., no effects) (Figure 5 and Figure 6B). Modeling further suggests that anti-spike and anti-nucleocapsid IgG dynamics are similar, except for their baseline value (*p* = 0.02, Wilcoxon rank-sum test) which could be due to differences in different assays that were used to measure them. The time difference between viral load take-off and the time of production of IgG was not statistically different between anti-spike (median = 6 days) and anti-nucleocapsid IgG (median = 13 days) (*p* = 0.29, Wilcoxon rank-sum test). Similarly, we found that rate of antibody production was higher for anti-nucleocapsid (median = 57.5/day) than anti-spike IgG (median = 10.9/day) but not statistically different (*p* = 0.14, Wilcoxon rank-sum test).

## 4. Discussion

Potential mitigating factors of COVID-19 severity include low inoculum dose and effective and early immune responses. However, the components of the immune response responsible for the elimination of SARS-CoV-2 replication are only partially understood. Early data suggests the high importance of an effective innate response during the first several days of infection [25,31,32,33]. Beyond this stage, effective acquired immune responses are likely critical [34,35,36,37,38,39,40]. Humoral immunity is thought to drive the high observed efficacy of two mRNA vaccines [41]. Therapeutic infusion of neutralizing antibodies early during infection is associated with more rapid elimination of viral shedding as well as decreased hospitalization [8,10,42,43].

Some studies have shown the association between anti-SARS-CoV-2 IgG and reduced viral loads, and a possible correlation between IgG concentration to neutralization titers [16,44]. Our modeling suggests that circulating antibodies play a limited role in eliminating nasal viral replication during the first exposure to the virus. This emerges from our modeling because IgG levels surge only well after viral loads have already started to decrease. The inclusion of a density-dependent killing term to capture peak viral load in our model already points to the likely importance of innate responses such as type I interferon responses [25]. There is ample evidence that the lack of type I interferon responses is associated with severe COVID-19 [31,32,45,46]. Studies have further shown both IFN-I pre-treatments of SARS-CoV-2 infection lead to 2-log10 to 4-log10 viral loads reduction [47,48,49], thus supporting the likely importance of innate response in early viral clearance.

SARS-CoV-2 specific antibodies exhibit both binding and neutralization activities in vitro [50,51]; however, their exact role during primary infection in vivo still remains unanswered [52]. Our analysis suggests that IgG anti-spike and anti-nucleocapsid antibodies may exhibit no effects on SARS-CoV-2 viral loads and that mild, late binding effects are more likely than neutralization effects to enhance viral clearance rate during primary infection; however, this effect is likely to be limited because the peak viral load occurs before the production of IgG starts and IgG antibodies only reach a high enough level at a relatively late stage of infection when viral loads are already at very low levels. Even in the absence of IgG antibodies, the kinetics of SARS-CoV-2 are not predicted to change dramatically. The modeling analysis also suggests that the generation of SARS-CoV-2 specific antibodies is akin to the generation of antibodies using traditional vaccines, such as in the case of vaccination for hepatitis B virus [53]. In both scenarios, B cell stimulation most likely occurs “once” in response to the viral antigen right after infection/vaccination leading to the programmed production of antibodies for months or years rather than continuous stimulation of naïve B cells in response to waning and waxing of viral antigen. Our study further suggests IgG antibody peak value is nearly significantly higher among severe patients whereas the difference for IgM peak value is smaller. Patients with different symptom severities react to SARS-CoV-2 infection with different antibody kinetics especially for IgG [30,31,32], which is mainly attributed to the generation rate of these antibodies, probably due to higher mean viral load in severe cases [54] and not to the timing of the production (relative to the onset of symptoms) or the coefficient that allows for the saturation of the production of these antibodies at high levels. No significant differences in the antibody kinetics of anti-spike and anti-nucleocapsid IgG were found but that could also be because of low sample sizes in our study.

While our results are only relevant for the first several weeks after infection, experimental evidence suggests that antibody titers are relatively stable for at least a period of 5 months after symptoms onset [2,3,55], with only ~3 fold decrease in antibodies levels in ~80–100 days [2,3]. This suggests that antibody production does not halt completely once SARS-CoV-2 viral loads go below the detection limit, which is typical following many viral infections.

Although IgG effects on SARS-CoV-2 infection might not be significant during primary infection, they are likely to be critical for preventing re-infection or following vaccination. For example, almost all individuals who were re-infected exhibited similar neutralization antibody (NAb) titers and only developed mild to moderate symptoms during the second infection [56]. SARS-CoV-2 rechallenge experiments with rhesus macaques also suggest that the neutralizing antibodies generated during the first time of infection may confer protective immunity against reinfection [57,58]. However, the protective immunity during re-infection could be linked to more than just the presence of IgG antibodies at the time of re-infection. For example, an analysis of a COVID-19 outbreak on a fishing vessel found that only those with potent neutralization but not just the presence of IgG antibody, are immune from reinfection [59]. Therefore, even if IgG antibodies may not play such a big part in clearing viruses during the primary infection, they might form effective protection against re-infection, and IgG levels can be an important indicator of the longevity of immunity.

There are important limitations in our study that limit the scope of our findings to hypothesis generation. First, our model could not confirm that nasal SARS-CoV-2 viral loads are not significantly affected by other humoral immune responses [60], which are not included in our study. For example, IgA are detected as early as only 1 day after symptom onset [50], which is long before IgG is detected but the influence of IgA on SARS-CoV-2 viral loads could not be confirmed due to the absence of longitudinal data and lack of measurement at the actual site of viral replication. Second, we did not model the alternative hypothesis that drives viral containment, as we had no data to validate or refute this type of model. The potentially most relevant T cells, those which are retained in tissue at the site of infection, have yet to be assessed for SARS-CoV-2. Third, we did not model, again for the lack of data, viral or antibody levels in the most relevant site of infection, the lung. It is unclear whether different kinetics underlie the clearance of virus from the lower airways [61]. Our results pertain more to viral loads that are relevant for transmission rather than pathogenesis. Fourth, our model was validated using a small number of patients, which could be addressed in future work by gathering more datasets. Finally, although parameter-identifiability issues in the best model were avoided by fixing several parameters in the fitting procedure and their non-existence is confirmed by univariate sensitivity analysis along with low values of the relative standard error of the estimated parameters, the same could not be performed for several other models given the lack of data. This again could be addressed in future work by gathering rich datasets.

In summary, we conclude that antibody responses to SARS-CoV-2 do not appear to be the primary mechanism underlying viral clearance from the nasal passages. Future studies should gather concurrent viral load, antibody, and T cell response data to better understand the importance of acquired immune responses in eliminating SARS-CoV-2.

## Figures and Tables

**Figure 1 viruses-13-00516-f001:**
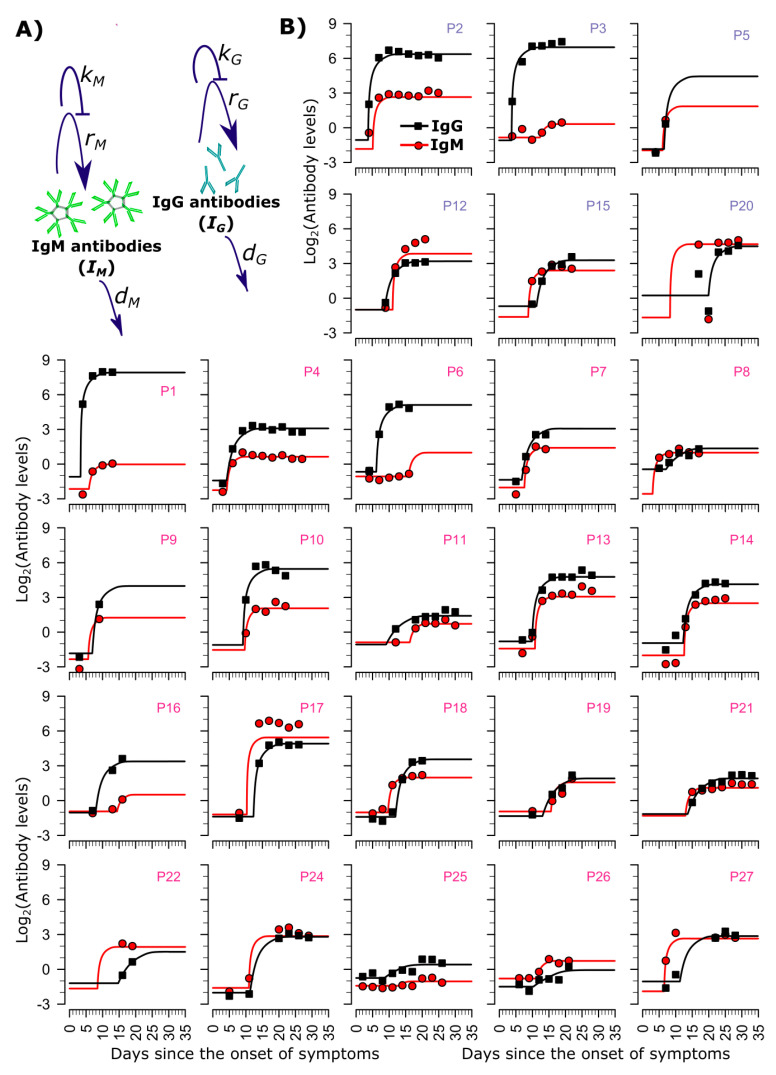
The mathematical model fits SARS-CoV-2 IgG and IgM levels following infection. (**A**) Schematic representation of the mathematical model reproducing longitudinal IgM and IgG dynamics, where $r_{M}$ and $r_{G}$ stand for the antibody production rates, $k_{M}$ and $k_{G}$ regulate the saturation of antibody generation, $d_{M}$ and $d_{G}$ are the natural clearance rates of IgM and IgG, respectively. The parameters used here correspond to the parameters in Equation (3) from the Materials and Methods section (or, model M3 in Appendix A), (**B**) Simulations (line), and data (markers) of IgM (red) and IgG (black) under the best model (model M3 in Appendix A). Severe and non-severe patients are labeled in blue and pink, respectively.

**Figure 2 viruses-13-00516-f002:**
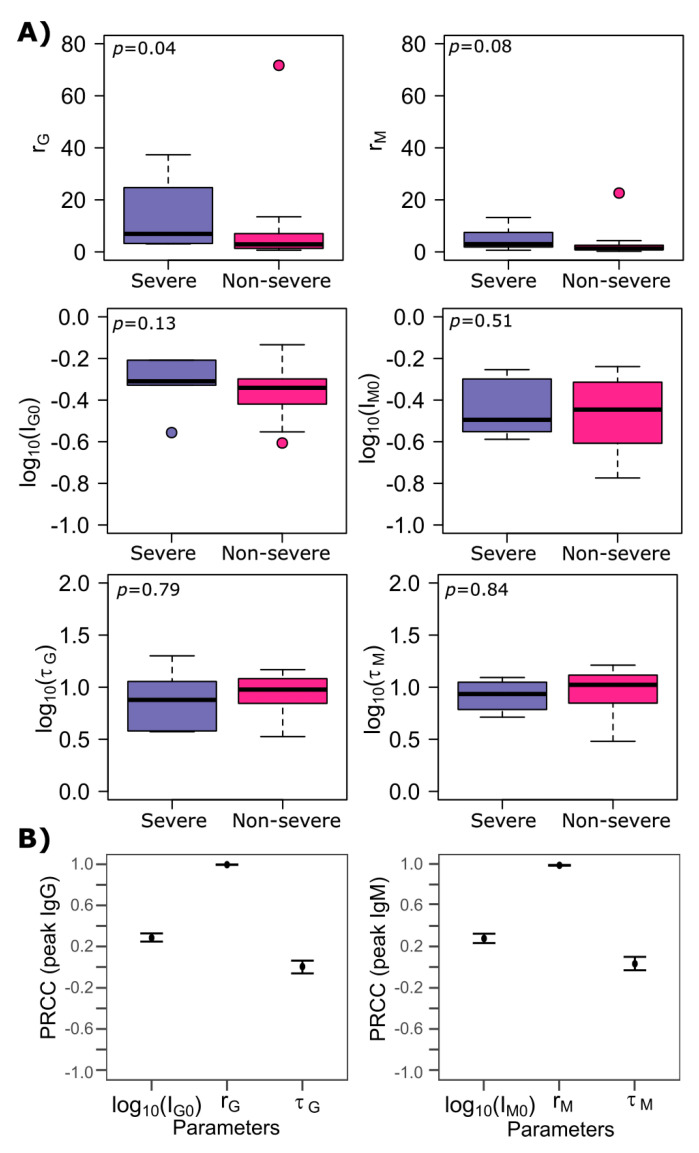
Model parameter comparison between non-severe and severe SARS-CoV-2 cases. (**A**) Comparison of estimated parameters between severe and non-severe patients using Wilcoxon-rank sum test under the model that best recapitulated the longitudinal IgM and IgG data of 6 severe and 20 non-severe-patients. We compared parameters $r_{M}$, $\log_{10}\left(I_{M0}\right)$, $r_{G}$, $\log_{10}\left(I_{G0}\right)$, $\log_{10}\left(\tau_{G}\right)$ and $\log_{10}\left(\tau_{G}\right)$ that represent the production rate of IgM, the log-converted concentration of IgM at *t* = 0, the production rate of IgG, the log-converted concentration of IgG at *t* = 0, time-delay (since the onset of symptoms) before IgG is produced and time-delay (since the onset of symptoms) before IgM is produced, respectively. *p* < 0.05 represents a statistically significant difference between severe and non-severe patients. (**B**) Using partial rank correlation coefficient, the sensitivity of the peak IgM and IgG levels to the initial antibody concentration, the production rate, and the delay before the production is induced.

**Figure 3 viruses-13-00516-f003:**
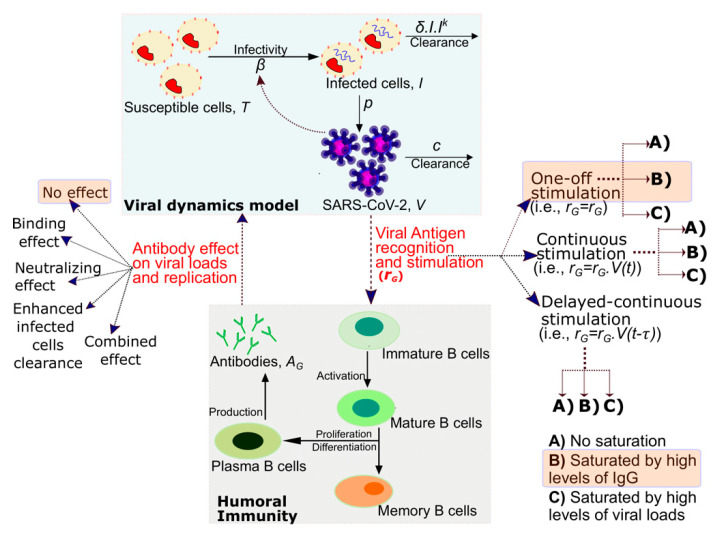
Mathematical model schematic of SARS-CoV-2 antibody generation and antiviral activity. The SARS-CoV-2 viral replication and antibody generation are displayed as two compartments. The SARS-CoV-2 viral antigen may trigger antibody immune response via one of three mechanisms: one-off stimulation, continuous stimulation, or delayed-continuous stimulation. For each mechanism, the antibody generation rate may proceed in 3 ways: no saturation, saturation by high levels of IgG, or saturation by viral load. The generation of antibodies may affect viral replication through viral binding effects, viral neutralization effects, via both mechanisms, or not at all. Two additional models containing antibody-assisted infected cell clearance are employed. In total, these multiple options give rise to 38 models. The components of the best-performing model are highlighted in orange rectangle boxes.

**Figure 4 viruses-13-00516-f004:**
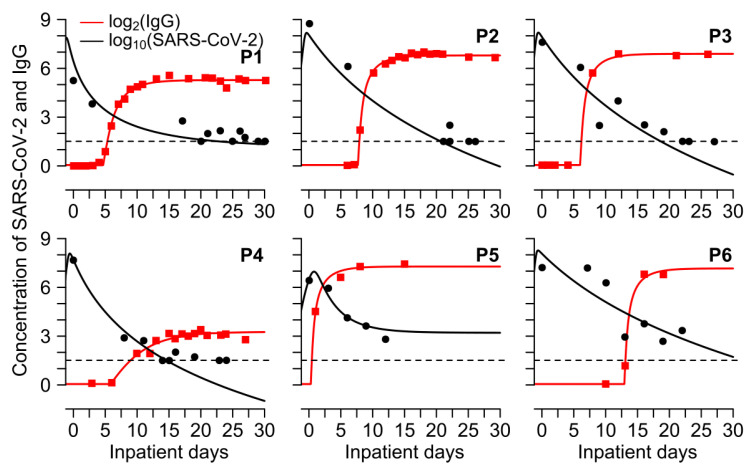
The mathematical model fits viral load and antibody levels following SARS-CoV-2 infection. Simulations (lines) to observed SARS-CoV-2 (black markers) and IgG (red markers) under the best model (MP-v2 in Appendix A).

**Figure 5 viruses-13-00516-f005:**
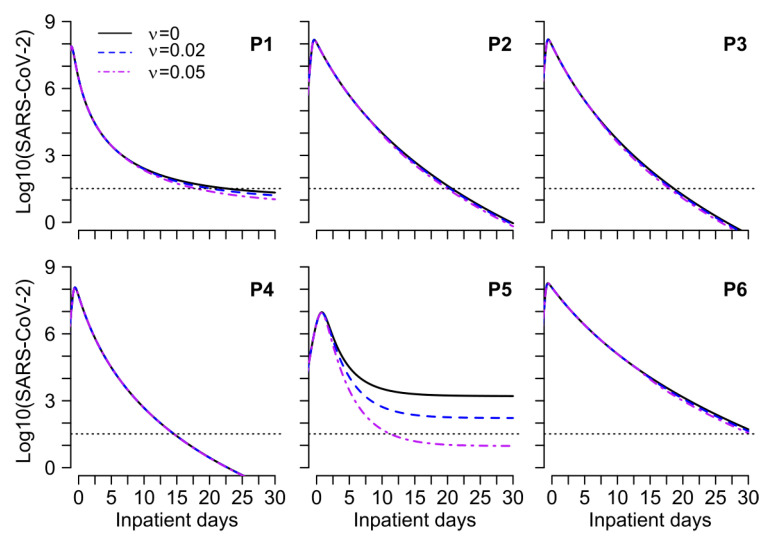
The slight impact of binding effects of nucleocapsid IgG antibodies on nasal viral dynamics. The solid black line represents the best fits to the observed data (using parameter values in Appendix A estimated under model MP-v2 in Appendix A), the dashed blue line representing the case when we assume weak binding effects (ν=0.02), and the dashed-dotted purple line represents strong binding effects (ν=0.05).

**Figure 6 viruses-13-00516-f006:**
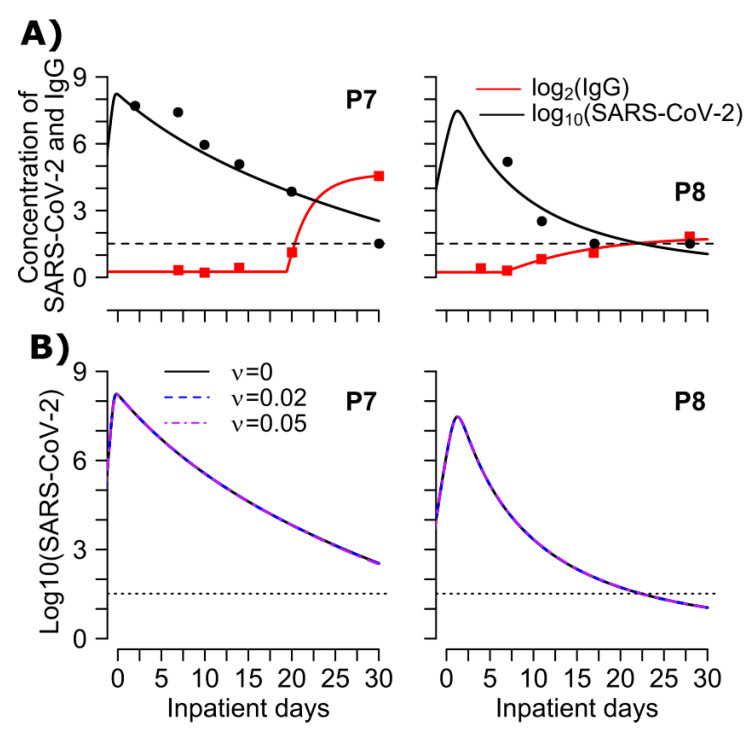
(**A**) Mathematical model fits viral load and antibody levels following SARS-CoV-2 infection. Simulations (lines) to observed SARS-CoV-2 (black markers) and anti-spike IgG (red markers) under the best model (MP-v2 in Appendix A). (B) Impact of binding effects of spike IgG antibodies on viral dynamics. The solid black line represents the best fits to the observed data (using parameter values in Appendix A estimated under model MP-v2 in Appendix A), the dashed blue line representing the case when we assume weak binding effects (ν=0.02), and the dashed-dotted purple line represents an enhanced strong binding efficacy of IgG antibodies (ν=0.05). The lines all notably overlap.

## Data Availability

Data is contained within the article or Appendix A.

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
