# Peer review of "Endogenously Produced SARS-CoV-2 Specific IgG Antibodies May Have a Limited Impact on Clearing Nasal Shedding of Virus during Primary Infection in Humans"

_viruses, 2021, doi:10.3390/v13030516_

Round 1
Reviewer 1 Report
Please see the attached pdf file.

Reviewer 2 Report
Manuscript by Yang et al. addresses the question of impact of SARS-CoV-2-specific IgG antibodies on clearing nasal virus during primary infection. The authors compared 36 different models to explore the interplay between virus and IgG antibodies using different mechanisms of antibody generation and the effect of antibodies on the virus neutralization and clearance.
Several points (especially point 2 below) need to be addressed before the publication.
1. The derivation of the model (3) used to fit the data in Figure 1 is confusing. Please write down the formulas for the transition from constants in models (1)/(2) to model (3) constants.
I think you can simply start directly from the model (3) by introducing it as a phenomenological model that captures the antibody kinetics in proposed way that is already nicely described in lines 117-127. The reason for this suggestion is because later on you will be using additional modifications of the equation for IgG kinetics (e.g. models MP-v2 vs MP-v3 vs MP-v6).
2. The parameters used for the models shown in the Table S3 lead to infection of all target cells during the first day of virus introduction and therefore are in a non-physiological part of the parameter space. For example, using reported parameters for the best model MQ-v2 and ID=1 (Table S4, see parameters and initial conditions listed below), all target cells T will be infected within 1 day from tzero (actual t=-10 days), and these infected cells will be further cleared by innate system (due to the term “delta*I*I^k”) leaving only few infected cells (less than 0.01% of initial number of target cells) at t=0. The concentration of virus V will be extremely high at the end of the first day of virus introduction. Please plot the dynamics of the model variables T, I and V starting from the day "tzero" (Table S4) to see it.
Unrealistically high initial level of target cell infection needs to be addressed.
Reported parameters: c=15, k=0.09, beta=5.9x10^-8, log10(p)=4.84 -> p=10^4.84=6.9*10^4, delta=0.4, nu=2.4, tzero= -10.27days
Initial conditions: T(tzero)=10^7, I(tzero)=1, V(tzero)=pI(tzero)/c=4612
Also, please explain the assumptions behind the term “delta*I*I^k”.
2. Figure 5 compares effect of IgG on nasal dynamics using modifications of model MQ-v2. By assuming no binding effects, the model MQ-v2 is transformed to the model MP-v2, and you already have shown in the Table S3 that it has slightly worth AIC. Is dashed blue line showing the results of the model MP-v2?
Minor comments:
- Schematic on Figure 3 shows constants (“pi” and “gamma”) that are not used in the equations.
- Authors compare “binding effects” and “neutralizing effects” of IgG on virus. Suggestion to replace “binding effects” with word “clearance” as for neutralization virus still needs to bind.
- Title of Figure 6 is confusing as you are plotting, I believe, only anti-spike IgG on that figure
- line 295 has the wrong reference
Reviewer 3 Report
The authors present a mathematical model for the endogenous production of SARS-CoV-2 specific antibodies and fit their parameters on clinical data from infected patients. As a general comment I find the paper really interesting and the models relatively well explained. I have just some comments that I think could help to further clarify its presentations.
- Please report the exact p-value of IgM and IgG levels in severe/non severe patients, you stated only that they satisfy p-value <0.05. From the pictures I am not totally convinced that there is a difference between severe/non severe patient antibodies levels.
- Authors could rearrange for clarity figure 1. Divide it in two, put in subfigure (a) all severe and (b) all non-severe patients. Now they are scattering around and is more difficult to see differences.
- Last line of equation 4 is the derivative with respect to T or t?
- Author test their model with three different set of IgG and IgM data. How do the different methods of measurement impact on the final values of parameters ?
- In the explanation of the fitting procedure it is not mentioned that the authors will fit the parameter patient by patient. If it is clear that the immune system can vary from patient to patient, in doing such a procedure there is the risk to lose the predictability of the model. Can you comment on this point? What happens if you fit only two set of parameters for a subset of severe and non-severe patients and you try to predict the remaining patients?
- IgG production is questionably significantly larger in severe than in non-severe with a p-value of 0.04 so authors could smooth their sentence. Assuming that this is true could the author comment on the biological consequence of that and why there is no difference instead in IgM levels?
- Do you choose to present the results of one model, what about the other ones? They are only very poorly commented. How the conclusions changed in your analysis if another method is considered? For example, how does Table S2 change?
- Line 305 you said that there is a high correlation. You report p = 0.92 that is misleading since I thought to p-value. Put r in instead of p.
- How do the model respond to the addition of non-endogenous antibodies for example from convalescent plasma or Regeneron/Lyly neutralizing Abs?
Round 2
Reviewer 1 Report
Thank you for addressing my concerns/comments reasonably. I don't have any further comments.
Reviewer 2 Report
The authors addressed my comments.
There are some typos from changing the text, for examples,
- before the best model was MQ-v2 and now MV-p2 is the best model, but the Figure 6 legend still partly use MQ-v2.
- The added fixed effect value “log10beta=7.23 virions-1day-1” seems to be missing “-“ as value “7.23” would be too high
Reviewer 3 Report
The authors addressed all the points that I raised up. I think that now the paper is ready for publication.